# Von Mises-Fisher Loss for Training Sequence to Sequence Models with Continuous Outputs

**Sachin Kumar & Yulia Tsvetkov**
Language Technologies Institute
Carnegie Mellon University
{sachink,ytsvetko}@cs.cmu.edu

## Abstract

The Softmax function is used in the final layer of nearly all existing sequence-to-sequence models for language generation. However, it is usually the slowest layer to compute which limits the vocabulary size to a subset of most frequent types; and it has a large memory footprint. We propose a general technique for replacing the softmax layer with a continuous embedding layer. Our primary innovations are a novel probabilistic loss, and a training and inference procedure in which we generate a probability distribution over pre-trained word embeddings, instead of a multinomial distribution over the vocabulary obtained via softmax. We evaluate this new class of sequence-to-sequence models with continuous outputs on the task of neural machine translation. We show that our models train up to 2.5x faster than the state-of-the-art models while achieving comparable translation quality. These models are capable of handling very large vocabularies without compromising on translation quality or speed. They also produce more meaningful errors than the softmax-based models, as these errors typically lie in a subspace of the vector space of the reference translations[1].

## 1 Introduction

Due to the power law distribution of word frequencies, rare words are extremely common in any language (Zipf, 1935). Yet, the majority of language generation tasks—including machine translation (Sutskever et al., 2014; Bahdanau et al., 2014; Luong et al., 2015), summarization (Rush et al., 2015; See et al., 2017; Paulus et al., 2018), dialogue generation (Vinyals & Le, 2015), question answering (Yin et al., 2015), speech recognition (Graves et al., 2013; Xiong et al., 2017), and others—generate words by sampling from a multinomial distribution over a closed output vocabulary. This is done by computing scores for each candidate word and normalizing them to probabilities using a *softmax* layer.

Since softmax is computationally expensive, current systems limit their output vocabulary to a few tens of thousands of most frequent words, sacrificing linguistic diversity by replacing the long tail of rare words by the unknown word token, ⟨unk⟩. Unsurprisingly, at test time this leads to an inferior performance when generating rare or out-of-vocabulary words. Despite the fixed output vocabulary, softmax is computationally the slowest layer. Moreover, its computation follows a large matrix multiplication to compute scores over the candidate words; this makes softmax expensive in terms of memory requirements and the number of parameters to learn (Mnih & Kavukcuoglu, 2013; Morin & Bengio, 2005; de Brébisson & Vincent, 2016). Several alternatives have been proposed for alleviating these problems, including sampling-based approximations of the softmax function (Bengio & Senecal, 2003; Mnih & Kavukcuoglu, 2013), approaches proposing a hierarchical structure of the softmax layer (Morin & Bengio, 2005; Chen et al., 2016), and changing the vocabulary to frequent subword units, thereby reducing the vocabulary size (Sennrich et al., 2016).

We propose a novel technique to generate low-dimensional continuous word representations, or word embeddings (Mikolov et al., 2013; Pennington et al., 2014; Bojanowski et al., 2017) instead of a probability distribution over the vocabulary at each output step. We train sequence-to-sequence

---

[1] The code is at https://github.com/Sachin19/seq2seq-con

models with continuous outputs by minimizing the distance between the output vector and the pre-trained word embedding of the reference word. At test time, the model generates a vector and then searches for its nearest neighbor in the target embedding space to generate the corresponding word. This general architecture can in principle be used for any language generation (or any recurrent regression) task. In this work, we experiment with neural machine translation, implemented using recurrent sequence-to-sequence models (Sutskever et al., 2014) with attention (Bahdanau et al., 2014; Luong et al., 2015).

To the best of our knowledge, this is the first work that uses word embeddings—rather than the softmax layer—as outputs in language generation tasks. While this idea is simple and intuitive, in practice, it does not yield competitive performance with standard regression losses like $\ell_2$. This is because $\ell_2$ loss implicitly assumes a Gaussian distribution of the output space which is likely false for embeddings. In order to correctly predict the outputs corresponding to new inputs, we must model the correct probability distribution of the target vector conditioned on the input (Bishop, 1994). A major contribution of this work is a new loss function based on defining such a probability distribution over the word embedding space and minimizing its negative log likelihood (§3).

We evaluate our proposed model with the new loss function on the task of machine translation, including on datasets with huge vocabulary sizes, in two language pairs, and in two data domains (§4). In §5 we show that our models can be trained up to 2.5x faster than softmax-based models while performing on par with state-of-the-art systems in terms of generation quality. Error analysis (§6) reveals that the models with continuous outputs are better at correctly generating rare words and make errors that are close to the reference texts in the embedding space and are often semantically-related to the reference translation.

## 2 BACKGROUND

Traditionally, all sequence to sequence language generation models use one-hot representations for each word in the output vocabulary $\mathcal{V}$. More formally, each word $w$ is represented as a unique vector $\mathbf{o}(w) \in \{0, 1\}^V$, where $V$ is the size of the output vocabulary and only one entry $id(w)$ (corresponding the word ID of $w$ in the vocabulary) in $\mathbf{o}(w)$ is 1 and the rest are set to 0. The models produce a distribution $\mathbf{p}_t$ over the output vocabulary at every step $t$ using the softmax function:

$$\mathbf{p}_t(w) = \frac{e^{s_w}}{\sum_{v \in \mathcal{V}} e^{s_v}} \tag{1}$$

where, $s_w = W_{hw}\mathbf{h}_t + b_w$ is the score of the word $w$ given the hidden state $\mathbf{h}$ produced by the LSTM cell (Hochreiter & Schmidhuber, 1997) at time step $t$. $W \in \mathbb{R}^{V \times H}$ and $b \in \mathbb{R}^v$ are trainable parameters. $H$ is the size of the hidden layer $\mathbf{h}$.

These parameters are trained by minimizing the negative log-likelihood (aka cross-entropy) of this distribution by treating $\mathbf{o}(w)$ as the target distribution. The loss function is defined as follows:

$$\text{NLL}(\mathbf{p_t}, \mathbf{o}(w)) = -\log(\mathbf{p_t}(w))$$

This loss computation involves a normalization proportional to the size of the output vocabulary $V$. This becomes a bottleneck in natural language generation tasks where the vocabulary size is typically tens of thousands of words. We propose to address this bottleneck by representing words as continuous word vectors instead of one-hot representations and introducing a novel probabilistic loss to train these models as described in §3.[2] Here, we briefly summarize prior work that aimed at alleviating the sofmax bottleneck problem.

---

[2]There is prior work on predicting word embeddings, but not in conditional language generation with seq2seq. Given a word embedding dictionary, Pinter et al. (2017) train a character-level neural net that learns to approximate the embeddings. It can then be applied to infer embeddings in the same space for words that were not available in the original set. These models were trained using the $\ell_2$ loss.

## 2.1 SOFTMAX ALTERNATIVES

We briefly summarize existing modifications to the sofmax layer, capitalizing on conceptually different approaches.

**Sampling-Based Approximations**  Sampling-based approaches completely do away with computing the normalization term of softmax by considering only a small subset of possible outputs. These include approximations like Importance Sampling (Bengio & Senecal, 2003), Noise Constrastive Estimation (Mnih & Kavukcuoglu, 2013), Negative Sampling (Mikolov et al., 2013), and Blackout (Ji et al., 2015). These alternatives significantly speed-up training time but degrade generation quality.

**Structural Approximations**  Morin & Bengio (2005) replace the flat softmax layer with a hierarchical layer in the form of a binary tree where words are at the leaves. This alleviates the problem of expensive normalization, but these gains are only obtained at training time. At test time, the hierarchical approximations lead to a drop in performance compared to softmax both in time efficiency and in accuracy. Chen et al. (2016) propose to divide the vocabulary into clusters based on their frequencies. Each word is produced by a different part of the hidden layer making the output embedding matrix much sparser. This leads to performance improvement both in training and decoding. However, it assigns fewer parameters to rare words which leads to inferior performance in predicting them (Ruder, 2017).

**Self Normalization Approaches**  Andreas et al. (2015); Devlin et al. (2014) add additional terms to the training loss which makes the normalization factor close to 1, obviating the need to explicitly normalize. The evaluation of certain words can be done much faster than in softmax based models which is extremely useful for tasks like language modeling. However, for generation tasks, it is necessary to ensure that the normalization factor is exactly 1 which might not always be the case, and thus it might require explicit normalization.

**Subword-Based Methods**  Jozefowicz et al. (2016) introduce character-based methods to reduce vocabulary size. While character-based models lead to significant decrease in vocabulary size, they often differentiate poorly between similarly spelled words with different meanings. Sennrich et al. (2016) find a middle ground between characters and words based on sub-word units obtained using Byte Pair Encoding (BPE). Despite its limitations (Oda et al., 2017), BPE achieves good performance while also making the model truly open vocabulary. BPE is the state-of-the art approach currently used in machine translation. We thus use this as a baseline in our experiments.

## 3   LANGUAGE GENERATION WITH CONTINUOUS OUTPUTS

In our proposed model, each word type in the output vocabulary is represented by a continuous vector $\mathbf{e}(w) \in \mathbb{R}^m$ where $m \ll V$. This representation can be obtained by training a word embedding model on a large monolingual corpus (Mikolov et al., 2013; Pennington et al., 2014; Bojanowski et al., 2017).

At each generation step, the decoder of our model produces a continuous vector $\hat{\mathbf{e}} \in \mathbb{R}^m$. The output word is then predicted by searching for the nearest neighbor of $\hat{\mathbf{e}}$ in the embedding space:

$$w_{\text{predicted}} = \operatorname*{argmin}_{w} \{d(\hat{\mathbf{e}}, \mathbf{e}(w)) | w \in \mathcal{V}\}$$

where $\mathcal{V}$ is the output vocabulary, $d$ is a distance function. In other words, the embedding space could be considered to be quantized into $V$ components and the generated continuous vector is mapped to a word based on the quanta in which it lies. The mapped word is then passed to the next step of the decoder (Gray, 1990). While training this model, we know the target vector $\mathbf{e}(w)$, and minimize its distance from the output vector $\hat{\mathbf{e}}$. With this formulation, our model is directly trained to optimize towards the information encoded by the embeddings. For example, if the embeddings are primarily semantic, as in Mikolov et al. (2013) or Bojanowski et al. (2017), the model would tend to output words in a semantic space, that is produced words would either be correct or close

synonyms (which we see in our analysis in §6), or if we use synactico-semantic embeddings (Levy & Goldberg, 2014; Ling et al., 2015), we might be able to also control for syntatic forms.

We propose a novel probabilistic loss function—a probabilistic variant of cosine loss—which gives a theoretically grounded regression loss for sequence generation and addresses the limitations of existing empirical losses (described in §4.2). Cosine loss measures the closeness between vector directions. A natural choice for estimating *directional* distributions is **von Mises-Fisher** (vMF) defined over a hypersphere of unit norm. That is, a vector close to the mean direction will have high probability. VMF is considered the directional equivalent of Gaussian distribution [3]. Given a target word $w$, its density function is given as follows:

$$p(\mathbf{e}(w); \boldsymbol{\mu}, \kappa) = C_m(\kappa) e^{\kappa \boldsymbol{\mu}^T \mathbf{e}(w)},$$

where $\boldsymbol{\mu}$ and $\mathbf{e}(w)$ are vectors of dimension $m$ with unit norm, $\kappa$ is a positive scalar, also called the concentration parameter. $\kappa = 0$ defines a uniform distribution over the hypersphere and $\kappa = \infty$ defines a point distribution at $\boldsymbol{\mu}$. $C_m(\kappa)$ is the normalization term:

$$C_m(\kappa) = \frac{\kappa^{m/2-1}}{(2\pi)^{m/2} I_{m/2-1}(\kappa)},$$

where $I_v$ is called modified Bessel function of the first kind of order $v$. The output of the model at each step is a vector $\hat{\mathbf{e}}$ of dimension $m$. We use $\kappa = \|\hat{\mathbf{e}}\|$. Thus the density function becomes:

$$p(\mathbf{e}(w); \hat{\mathbf{e}}) = \text{vMF}(\mathbf{e}(w); \hat{\mathbf{e}}) = C_m(\|\hat{\mathbf{e}}\|) e^{\hat{\mathbf{e}}^T \mathbf{e}(w)} \tag{2}$$

It is noteworthy that equation 2 is very similar to softmax computation (except that $\mathbf{e}(\mathbf{w})$ is a unit vector), the main difference being that normalization is not done by summing over the vocabulary, which makes it much faster than the softmax computation. More details about it's computation are given in the appendix.

The negative log-likelihood of the vMF distribution, which at each output step is given by:

$$\text{NLLvMF}(\hat{\mathbf{e}}; \mathbf{e}(w)) = -\log\left(C_m(\|\hat{\mathbf{e}}\|)\right) - \hat{\mathbf{e}}^T \mathbf{e}(w)$$

**Regularization of NLLvMF**  In practice, we observe that the NLLvMF loss puts too much weight on increasing $\|\hat{\mathbf{e}}\|$, making the second term in the loss function decrease rapidly without significant decrease in the cosine distance. To account for this, we add a regularization term. We experiment with two variants of regularization.

$\text{NLLvMF}_{\text{reg}_1}$: We add $\lambda_1 \|\hat{\mathbf{e}}\|$ to the loss function, where $\lambda_1$ is a scalar hyperparameter.[4]  This makes intuitive sense in that the length of the output vector should not increase too much. The regularized loss function is as follows:

$$\text{NLLvMF}_{\text{reg}_1}(\hat{\mathbf{e}}) = -\log C_m(\|\hat{\mathbf{e}}\|) - \hat{\mathbf{e}}^T \mathbf{e}(w) + \lambda_1 \|\hat{\mathbf{e}}\|$$

$\text{NLLvMF}_{\text{reg}_2}$: We modify the previous loss function as follows:

$$\text{NLLvMF}_{\text{reg}_2}(\hat{\mathbf{e}}) = -\log C_m(\|\hat{\mathbf{e}}\|) - \lambda_2 \hat{\mathbf{e}}^T \mathbf{e}(w) \tag{3}$$

$-\log C_m(\|\hat{\mathbf{e}}\|)$ decreases slowly as $\|\hat{\mathbf{e}}\|$ increases as compared the second term. Adding a $\lambda_2 < 1$ the second term controls for how fast it can decrease.[5]

## 4 EXPERIMENTS

### 4.1 EXPERIMENTAL SETUPS

We modify the standard seq2seq models in OpenNMT[6] in PyTorch[7] (Klein et al., 2017) to implement the architecture described in §3. This model has a bidirectional LSTM encoder with an attention-based decoder (Luong et al., 2015). The encoder has one layer whereas the decoder has 2 layers of

---

[3] A natural choice for many regression tasks would be to use a loss function based on Gaussian distribution itself which is a probabilistic version of $\ell_2$ loss. But as we describe in §4.2, $\ell_2$ is not considered a suitable loss for regression on embedding spaces

[4] We empirically set $\lambda_1 = 0.02$ in all our experiments

[5] We use $\lambda_2 = 0.1$ in all our experiments

[6] http://opennmt.net/

[7] https://pytorch.org/

size 1024 with the input word embedding size of 512. For the baseline systems, the output at each decoder step multiplies a weight matrix ($H \times V$) followed by softmax. This model is trained until convergence on the validation perplexity. For our proposed models, we replace the softmax layer with the continuous output layer ($H \times m$) where the outputs are $m$ dimensional. We empirically choose $m = 300$ for all our experiments. Additional hyperparameter settings can be found in the appendix. These models are trained until convergence on the validation loss. Out of vocabulary words are mapped to an ⟨unk⟩ token[8]. We assign ⟨unk⟩ an embedding equal to the average of embeddings of all the words which are not present in the target vocabulary of the training set but are present in vocabulary on which the word embeddings are trained. Following Denkowski & Neubig (2017), after decoding a post-processing step replaces the ⟨unk⟩ token using a dictionary look-up of the word with highest attention score. If the word does not exist in the dictionary, we back off to copying the source word itself. Bilingual dictionaries are automatically extracted from our parallel training corpus using word alignment (Dyer et al., 2013)[9]. We evaluate all the models on the test data using the BLEU score (Papineni et al., 2002).

We evaluate our systems on standard machine translation datasets from IWSLT'16 (Cettolo et al., 2016), on two target languages, English: German→English, French→English and a morphologically richer language French: English→French. The training sets for each of the language pairs contain around 220,000 parallel sentences. We use TED Test 2013+2014 (2,300 sentence pairs) as developments sets and TED Test 2015+2016 (2,200 sentence pairs) as test sets respectively for all the language pairs. All mentioned setups have a total vocabulary size of around 55,000 in the target language of which we choose top 50,000 words by frequency as the target vocabulary[10].

We also experiment with a much larger WMT'16 German→English (Bojar et al., 2016) task whose training set contains around 4.5M sentence pairs with the target vocabulary size of around 800,000. We use newstest2015 and newstest2016 as development and test data respectively. Since with continuous outputs we do not need to perform a time consuming softmax computation, we can train the proposed model with very large target vocabulary without any change in training time per batch. We perform this experiment with WMT'16 de–en dataset with a target vocabulary size of 300,000 (basically all the words in the target vocabulary for which we had trained embeddings). But to able to produce these words, the source vocabulary also needs to be increased to have their translations in the inputs, which would lead to a huge increase in the number of trainable parameters. Instead, we use sub-words computed using BPE as source vocabulary. We use 100,000 merge operations to compute the source vocabulary as we observe using a smaller number leads to too small (and less meaningful) sub-word units which are difficult to align with target words.

Both of these datasets contain examples from vastly different domains, while IWSLT'16 contains less formal spoken language, WMT'16 contains data primarily from news.

We train target word embeddings for English and French on corpora constructed using WMT'16 (Bojar et al., 2016) monolingual datasets containing data from Europarl, News Commentary, News Crawl from 2007 to 2015 and News Discussion (everything except Common Crawl due to its large memory requirements). These corpora consist of 4B+ tokens for English and 2B+ tokens for French. We experiment with two embedding models: word2vec Mikolov et al. (2013) and fasttext Bojanowski et al. (2017) which were trained using the hyper-parameters recommended by the authors.

## 4.2 EMPIRICAL LOSS FUNCTIONS

We compare our proposed loss function with standard loss functions used in multivariate regression.

**Squared Error** ($\ell_2$) is the most common distance function used when the model outputs are continuous (Lehmann & Casella, 1998). For each target word $w$, it is given as $\mathcal{L}_{\ell_2} = \|\hat{\mathbf{e}} - \mathbf{e}(w)\|^2$

$\ell_2$ penalizes large errors more strongly and therefore is sensitive to outliers. To avoid this we use a square rooted version of $\ell_2$ loss. But it has been argued that there is a mismatch between the objective function used to learn word representations (maximum likelihood based on inner product), the distance measure for word vectors (cosine similarity), and $\ell_2$ distance as the objective function

---

[8]Although the proposed model can make decoding open vocabulary, there could still be unknown words, e.g., words for which we do not have pre-trained embeddings; we need ⟨unk⟩ token to represent these words

[9]`https://github.com/clab/fast_align`

[10]Removing the bottom 5,000 words did not make a significant difference in terms of translation quality

to learn transformations of word vectors (Xing et al., 2015). This argument prompts us to look at cosine loss.

**Cosine Loss** is given as $\mathcal{L}_{\text{cosine}} = 1 - \frac{\hat{\mathbf{e}}^T \mathbf{e}(w)}{\|\hat{\mathbf{e}}\| . \|\mathbf{e}(w)\|}$. This loss minimizes the distance between the directions of output and target vectors while disregarding their magnitudes. The target embedding space in this case becomes a set of points on a hypersphere of dimension $m$ with unit radius.

**Max Margin Loss** Lazaridou et al. (2015) argue that using pairwise losses like $\ell_2$ or cosine distance for learning vectors in high dimensional spaces leads to *hubness*: word vectors of a subset of words appear as nearest neighbors of many points in the output vector space. To alleviate this, we experiment with a margin-based ranking loss (which has been shown to reduce hubness) to train the model to rank the word vector prediction $\hat{\mathbf{e}}$ for target vector $\mathbf{e}(w)$ higher than any other word vector $\mathbf{e}(w')$ in the embedding space. $\mathcal{L}_{\text{mm}} = \sum_{w' \in \mathcal{V}, w' \neq w} \max\{0, \gamma + \cos(\hat{\mathbf{e}}, \mathbf{e}(w')) - \cos(\hat{\mathbf{e}}, \mathbf{e}(w))\}$ where, $\gamma$ is a hyperparameter[11] representing the margin and $w'$ denotes negative examples. We use only one informative negative example as described in Lazaridou et al. (2015) which is closest to $\hat{\mathbf{e}}$ and farthest from the target word vector $\mathbf{e}(w)$. But, searching for this negative example requires iterating over the vocabulary which brings back the problem of slow loss computation.

### 4.3 DECODING

In the case of empirical losses, we output the word whose target embedding is the nearest neighbor to the vector in terms of the distance (loss) defined. In the case of NLLvMF, we predict the word whose target embedding has the highest value of vMF probability density wrt to the output vector. This predicted word is fed as the input for the next time step. Our nearest-neighbor decoding scheme is equivalent to a greedy decoding; we thus compare to baseline models with beam size of 1.

### 4.4 TYING THE TARGET EMBEDDINGS

Until now we discussed the embeddings in the output layer. Additionally, decoder in a sequence-to-sequence model has an *input* embedding matrix as the previous output word is fed as an input to the decoder. Much of the size of the trainable parameters in all the models is occupied by these input embedding weights. We experiment with keeping this embedding layer fixed and tied with pre-trained target output embeddings (Press & Wolf, 2016). This leads to significant reduction in the number of parameters in our model.

## 5 RESULTS

**Translation Quality** Table 1 shows the BLEU scores on the test sets for several baseline systems, and various configurations including the types of losses, types of inputs/outputs used (word, BPE, or embedding)[12] and whether the model used tied embeddings in the decoder or not.

$\ell_2$ loss attains the lowest BLEU scores among the proposed models; our manual error analysis reveals that the high error rate is due to the hubness phenomenon, as we described in §4.2. The BLEU scores improve for cosine loss, confirming the argument of Xing et al. (2015) that cosine distance is a better suited similarity (or distance) function for word embeddings. Best results—for MaxMargin and NLLvMF losses—surpass the strong BPE baseline in translation French→English and English→French, and attain slightly lower but competitive results on German→English.

Since we represent each target word by its embedding, the quality of embeddings should have an impact on the translation quality. We measure this by training our best model with fasttext embeddings (Bojanowski et al., 2017), which leads to $> 1$ BLEU improvement. Tied embeddings are the most effective setups: they not only achieve highest translation quality, but also dramatically reduce parameters requirements and the speed of convergence.

---

[11]We use $\gamma = 0.5$ in our experiments.

[12]Note that we do not experiment with subword embeddings since the number of merge operations for BPE usually depend on the choice of a language pair which would require the embeddings to be retrained for every language pair.

| Embedding Model | Tied Emb | Source Type/ Target Type | Loss | BLEU | | |
|---|---|---|---|---|---|---|
| | | | | fr–en | de–en | en–fr |
| - | no | word→word | CE | 31.0 | 24.7 | 29.3 |
| - | no | word→BPE | CE | 29.1 | 24.1 | 29.8 |
| - | no | BPE→BPE | CE | **31.4** | **25.8** | **31.0** |
| word2vec | no | word→emb | L2 | 27.2 | 19.4 | 26.4 |
| word2vec | no | word→emb | Cosine | 29.1 | 21.9 | 26.6 |
| word2vec | no | word→emb | MaxMargin | 29.6 | 21.4 | 26.7 |
| fasttext | no | word→emb | MaxMargin | 31.0 | 25.0 | 29.0 |
| fasttext | yes | word→emb | MaxMargin | **32.1** | **25.0** | **31.0** |
| word2vec | no | word→emb | $\text{NLLvMF}_{\text{reg1}}$ | 29.5 | 22.7 | 26.6 |
| word2vec | no | word→emb | $\text{NLLvMF}_{\text{reg1+reg2}}$ | 29.7 | 21.6 | 26.7 |
| word2vec | yes | word→emb | $\text{NLLvMF}_{\text{reg1+reg2}}$ | 29.7 | 22.2 | 27.5 |
| fasttext | no | word→emb | $\text{NLLvMF}_{\text{reg1+reg2}}$ | 30.4 | 23.4 | 27.6 |
| fasttext | yes | word→emb | $\text{NLLvMF}_{\text{reg1+reg2}}$ | **32.1** | **25.1** | **31.7** |

Table 1: Translation quality experiments (BLEU scores) on IWSLT16 datasets

Table 2 shows results on WMT'16 test set in terms of BLEU and METEOR (Denkowski & Lavie, 2014) trained only for best-performing setups in table 1. METEOR uses paraphrase tables and WordNet synonyms for common words. This may explain why METEOR scores, unlike BLEU, close the gap with the baseline models: as we found in the qualitative analysis of outputs, our models often output synonyms of the reference words, which are plausible translations but are penalized by BLEU. [13] Examples are included in the Appendix.

| Loss | BLEU | METEOR |
|---|---|---|
| CE | 22.9 | 23.9 |
| CE (BPE) | 30.1 | 28.7 |
| MaxMargin | 24.3 | 25.2 |
| $\text{NLLvMF}_{reg_1+reg_2}$ | 28.8 | 28.2 |

Table 2: Translation quality experiment on WMT16 de–en

**Training Time**   Table 4 shows the average training time per batch. In figure 1 (left), we show how many samples per second our proposed model can process at training time compared to the baseline. As we increase the batch size, the gap between the baseline and the proposed models increases. Our proposed models can process large mini-batches while still training much faster than the baseline models. The largest mini-batch size with which we can train our model is 512, compared to 184 in the baseline model. Using max-margin loss leads to a slight increase in the training time compared to NLLvMF. This is because its computation needs a negative example which requires iterating over the entire vocabulary. Since our model requires look-up of nearest neighbors in the target embedding table while testing, it currently takes similar time as that of softmax-based models. In future work, approximate nearest neighbors algorithms Johnson et al. (2017) can be used to improve translation time.

We also compare the speed of convergence, using BLEU scores on dev data. In figure 1 (right), we plot the BLEU scores against the number of epochs. Our model convergences much faster than the baseline models leading to an even larger improvement in overall training time (Similar figures for more datasets can be found in the appendix). As a result, as shown in table 3, the total training time of our proposed model (until convergence) is less than up-to 2.5x of the total training time of the baseline models.

**Memory Requirements**   As shown in Table 4 our best performing model requires less than 1% of the number of parameters in input and output layers, compared to BPE-based baselines.

---

[13] In IWSLT'16 datasets we obtain similar performances in BLEU and METEOR, this is likely because those models perform better particularly in translating rare words (§6) which are not covered in METEOR resources.

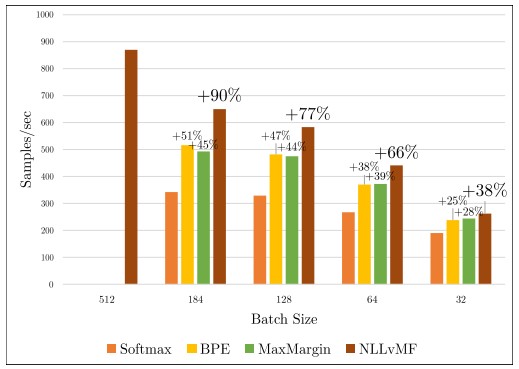 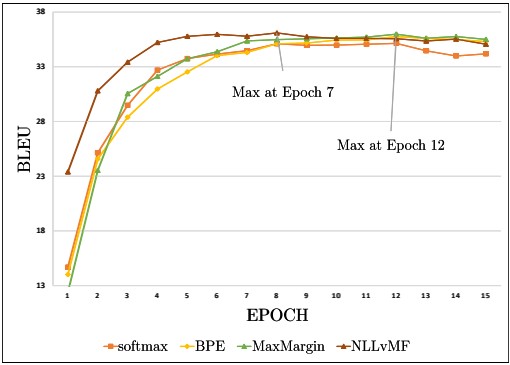

Figure 1: Left: Comparison of samples processed per second by the softmax vs. BPE vs. continuous output vMF models for IWSLT16 fr–en. Right: Comparison of convergence times of our models and baseline models on IWSLT16 fr–en validation sets. Baseline softmax as well as BPE converge at epoch 12 whereas our proposed model (NLLvMF) converges at epoch 7.

|  | **Softmax** | **BPE** | **Emb w/ NLL-vMF** |
|---|---|---|---|
| fr–en | 4h | 4.5h | **1.9h** |
| de–en | 3h | 3.5h | **1.5h** |
| en–fr | 1.8h | 2.8h | **1.3** |
| WMT de–en | 4.3d | 4.5d | **1.6d** |

Table 3: Total convergence times in hours(h)/days(d).

# 6 ERROR ANALYSIS

**Translation of Rare Words**   We evaluate the translation accuracy of words in the test set based on their frequency in the training corpus. Table 5 shows how the $F_1$ score varies with the word frequency. $F_1$ score gives a balance between recall (the fraction of words in the reference that the predicted sentence produces right) and precision (the fraction of produced words that are in reference). We show substantial improvements over softmax and BPE baselines in translating less frequent and rare words, which we hypothesize is due to having learned good embeddings of such words from the monolingual target corpus where these words are not as rare. Moreover, in BPE based models, rare words on the source side are split in smaller units which are in some cases not properly translated in subword units on the target side if transparent alignments don't exist. For example, the word *saboter* in French is translated to *sab+ot+tate* by the BPE model whereas correctly translated as *sabotage* by our model. Also, a rare word *retraite* in French in translated to *pension* by both Softmax and BPE models (pension is a related word but less rare in the corpus) instead of the expected translation *retirement* which our model gets right.

We conducted a thorough analysis of outputs across our experimental setups. Few examples are shown in the appendix. Interestingly, there are many examples where our models do not exactly match the reference translations (so they do not benefit from in terms of BLEU scores) but produce meaningful translations. This is likely because the model produces nearby words of the target words or paraphrases instead of the target word (which are many times synonyms).

Since we are predicting embeddings instead of actual words, the model tends to be weaker sometimes and does not follow a good language model and leads to ungrammatical outputs in cases where the baseline model would perform well. Integrating a pre-trained language model within the decoding framework is one potential avenue for our future work. Another reason for this type of errors could be our choice of target embeddings which are not modeled to (explicitly) capture syntactic relationships. Using syntactically inspired embeddings (Levy & Goldberg, 2014; Ling et al., 2015) might help reduce these errors. However, such fluency errors are not uncommon also in softmax and BPE-based models either.

| Output Type | Tied | Loss | #Parameters Input Layer | #Parameters Output Layer | Training time (ms) |
|---|---|---|---|---|---|
| word | No | CE | 25.6M (1.0x) | 51.2M (1.0x) | 400 (1.0x) |
| BPE | No | CE | 8.192M (0.32x) | 16.384M (0.32x) | 346 (0.86x) |
| emb | No | L2 | 25.6M (1.0x) | 307.2K (0.006x) | 160 (0.4x) |
| emb | No | Cosine | 25.6M (1.0x) | 307.2K (0.006x) | 160 (0.4x) |
| emb | No | MaxMargin | 25.6M (1.0x) | 307.2K (0.006x) | 178 (0.43x) |
| emb | Yes | MaxMargin | 153.6K (0.006x) | 307.2K (0.006x) | 178 (0.43x) |
| emb | No | $\text{NLLvMF}_x$ | 25.6M (1.0x) | 307.2K (0.006x) | 170 (0.42x) |
| emb | Yes | $\text{NLLvMF}_x$ | 153.6K (0.006x) | 307.2K (0.006x) | 170 (0.42x) |

Table 4: Comparison of number of parameters needed for input and output layer, train time per batch (with batch size of 64) for IWSLT16 fr–en. Numbers in parentheses indicate the fraction of parameters compared to word/word baseline model.

| Word Freq | Softmax | BPE | Max Margin | Emb w/ NLL-vMF |
|---|---|---|---|---|
| 1 | 0.42 | 0.50 | 0.30 | **0.52** |
| 2 | 0.16 | 0.26 | 0.25 | **0.31** |
| 3 | 0.14 | 0.22 | 0.25 | **0.33** |
| 4 | 0.29 | 0.24 | 0.30 | **0.33** |
| 5-9 | 0.28 | 0.33 | **0.38** | 0.37 |
| 10-99 | 0.54 | 0.53 | 0.53 | **0.55** |
| 100-999 | 0.60 | **0.61** | 0.60 | 0.60 |
| 1000+ | 0.69 | **0.70** | 0.69 | 0.69 |

Table 5: Test set unigram $F_1$ scores of occurrence in the predicted sentences based on their frequencies in the training corpus for different models for fr–en.

## 7  CONCLUSION

This work makes several contributions. We introduce a novel framework of sequence to sequence learning for language generation using word embeddings as outputs. We propose new probabilistic loss functions based on vMF distribution for learning in this framework. We then show that the proposed model trained on the task of machine translation leads to reduction in trainable parameters, to faster convergence, and a dramatic speed-up, up to 2.5x in training time over standard benchmarks. Table 6 visualizes a comparison between different types of softmax approximations and our proposed method.

State-of-the-art results in softmax-based models are highly optimized after a few years on research in neural machine translation. The results that we report are comparable or slightly lower than the strongest baselines, but these setups are only an initial investigation of translation with the continuous output layer. There are numerous possible directions to explore and improve the proposed setups. What are additional loss functions? How to setup beam search? Should we use scheduled sampling? What types of embeddings to use? How to translate with the embedding output into morphologically-rich languages? Can low-resource neural machine translation benefit from translation with continuous outputs if large monolingual corpora are available to pre-train strong target-side embeddings? We will explore these questions in future work.

Furthermore, the proposed architecture and the probabilistic loss (NLLvMF) have the potential to benefit other applications which have sequences as outputs, e.g. speech recognition. NLLvMF could be used as an objective function for problems which currently use cosine or $\ell_2$ distance, such as learning multilingual word embeddings. Since the outputs of our models are continuous (rather than class-based discrete symbols), these models can potentially simplify training of generative adversarial networks for language generation.

| | Sampling Based | Structure Based | Subword Units | Emb w/ NLL-vMF |
|---|:---:|:---:|:---:|:---:|
| **Training Time** | 🙂 | 🙂 | 🙂 | 🙂 |
| **Test Time** | 😐 | 😐 | 🙂 | 🙂 |
| **Accuracy** | 🙁 | 🙁 | 🙂 | 🙂 |
| **Parameters** | 😐 | 🙁 | 🙂 | 🙂 |
| **Handle Huge Vocab** | 🙁 | 🙁 | 🙂 | 🙂 |

Table 6: Comparison of softmax alternatives. Red denotes worse than softmax, green denotes better than softmax (fractional improvements) and blue denotes huge improvement (more than 2X) over softmax.

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

# 8 APPENDIX

## 8.1 HYPERPARAMETER AND INFRASTRUCTURE DETAILS

| Parameter | Value |
|---|---|
| LSTM Layers: Encoder | 1 |
| LSTM Layers: Decoder | 2 |
| Hidden Dimension (H) | 1024 |
| Input Word Embedding Size | 512 |
| Output Vector Size | 300 |
| Optimizer | Adam |
| Learning Rate (Baseline) | 0.0002 |
| Learning Rate (Our Models) | 0.0005 |
| Max Sentence Length | 100 |
| Vocabulary Size (Source) | 50000 |
| Vocabulary Size (Target) | 50000 |

Table 7: Hyperparameters Details

| | |
|---|---|
| PyTorch | 0.3.0 |
| CPU | Intel(R) Xeon(R) CPU 2.40GHz (32 Cores) |
| RAM | 190G |
| #GPUs/experiment | 1 |
| GPU | GeForce GTX TITAN X |

Table 8: Infrastructure details. All the experiments were run with this configuration

## 8.2 GRADIENT COMPUTATION FOR NLLvMF LOSS

NLLvMF loss is given as

$$\text{NLLvMF}(\hat{\mathbf{e}}; \mathbf{e}(w)) = -\log\left(C_m \|\hat{\mathbf{e}}\|\right) - \hat{\mathbf{e}}^T \mathbf{e}(w),$$

where $C_m(\kappa)$ is given as:

$$C_m(\kappa) = \frac{\kappa^{m/2-1}}{(2\pi)^{m/2} I_{m/2-1}(\kappa)}.$$

The normalization constant is not directly differentiable because Bessel function cannot be written in a closed form. The gradient of the first component ($\log\left(C_m \|\hat{\mathbf{e}}\|\right)$) of the loss is given as

$$\Delta \log(C_m(\kappa)) = -\frac{I_{m/2}(\kappa)}{I_{m/2-1}(\kappa)}.$$

This involves two computations of Bessel function $(I_v(z))$ for $m = 300$ for which we use `scipy.special.ive`. For high values of $v$[14] and low values of $z$, the values of the Bessel function can become really small and lead to underflow (but the gradient is still large). To deal with underflow, the gradient value could be approximated with it's (tight) lower bound (Ruiz-Antoln & Segura, 2016),

$$-\frac{I_{m/2}(\kappa)}{I_{m/2-1}(\kappa)} \geq -\frac{z}{v - 1 + \sqrt{(v+1)^2 + z^2}}$$

That is, in the initial steps of training, one might need to use to the approximation of the gradient to train the model and switch to the actual computation later on. One could also approximate the value of $\log(C_m(\kappa))$ by integrating over the approximate gradient value which is given as

$$\log(C_m(\kappa)) \geq \sqrt{(v+1)^2 + z^2} - (v - 1)\log(v - 1 + \sqrt{(v+1)^2 + z^2}).$$

In practice, we see that replacing $\log(C_m(\kappa))$ with this approximation in the loss function gives similar performance on the test data as well alleviates the problem of underflow. We thus recommend using it.

---

[14]for $m = 300$, we don't face this issue, but it is useful if one is using embeddings of higher dimensions

### 8.3 TRANSLATION QUALITY AND PERFORMANCE: ADDITIONAL RESULTS

Figure 2 shows the convergence time results for more IWSLT datasets. The results shown are averaged over multiple runs, and are in line with results reported in Figure 1.

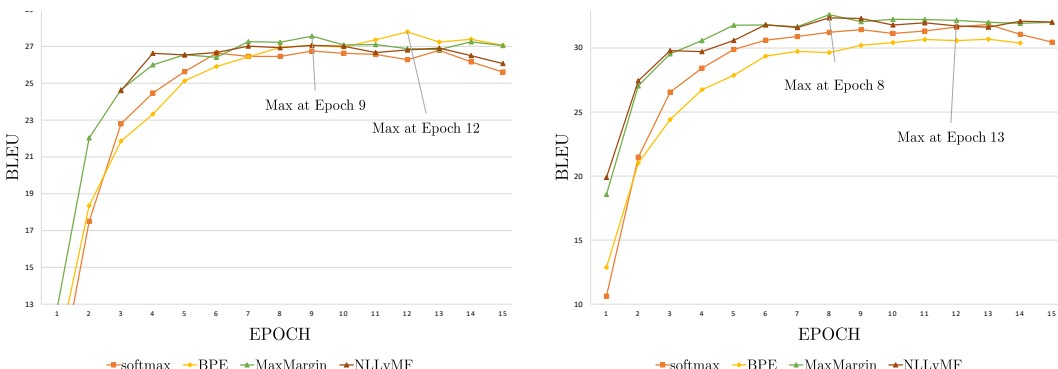

Figure 2: Comparison of convergence times of our models and baseline models on IWSLT16 de–en (left) and en–fr (right) validation sets.

In Table 1, we present results of translation quality with our proposed model and comparable baselines with a beam size of one. Here, for completeness, table 9 shows additional results with softmax-based models with a beam size of 5.

| Loss | BLEU |
|------------|------|
| IWSLT fr–en | 32.2 |
| IWSLT de–en | 26.1 |
| IWSLT en–fr | 32.4 |
| WMT de–en | 31.9 |

Table 9: Translation quality experiments using beam search with BPE based baseline models with a beam size of 5

With our proposed models, in principle, it is possible to generate candidates for beam search by using $K$-Nearest Neighbors. But how to rank the partially generated sequences is not trivial (one could use the loss values themselves to rank, but initial experiments with this setting did not result in significant gains). In this work, we focus on enabling training with continuous outputs efficiently and accurately giving us huge gains in training time. The question of decoding with beam search requires substantial investigation and we leave it for future work.

## 8.4 SAMPLE TRANSLATIONS FROM TEST SETS

| | |
|---|---|
| Input | Une ducation est critique, mais rgler ce problme va ncessiter que chacun d'entre nous s'engage et soit un meilleur exemple pour les femmes et filles dans nos vies. |
| Reference | An education is critical, but tackling this problem is going to require each and everyone of us to step up and be better role models for the women and girls in our own lives. |
| Predicted (BPE) | Education is critical, but it's going to require that each of us *will come in and if you do* a better **example** for women and girls in our lives. |
| Predicted (L2) | Education is critical , but to to do this is going to require that each of us *of* to engage and *or* a better **example** *of* the women and girls in our lives. |
| Predicted (Cosine) | That's critical , but *that's that it's* going to require that each of us is going to *take that* the problem and they're going *to if* you're a better **example** for women and girls in our lives. |
| Predicted (MaxMargin) | Education is critical, but that problem is going to require that every one of us **is engaging** and **is a better example** for women and girls in our lives. |
| Predicted (NLLvMF$_{reg}$) | Education is critical , but *fixed* this problem is going to require that **all of us engage** and **be a better example** for women and girls in our lives. |

Table 10: Translation examples. Red and blue colors highlight translation errors; red are bad and blue are outputs that are good translations, but are considered as errors by the BLEU metric. Our systems tend to generate a lot of such "meaningful" errors.

| | |
|---|---|
| Input | Pourquoi ne sommes nous pas de simples robots qui traitent toutes ces donnes, produisent ces rsultats, sans faire l'exprience de ce film intrieur ? |
| Reference | Why aren't we just robots who process all this input, produce all that output, without experiencing the inner movie at all? |
| Predicted (BPE) | Why *don't we have* simple robots that *are processing* all of this data, produce **these results**, without *doing* **the experience of that** inner movie? |
| Predicted (L2) | Why are we not *that we do that that are technologized and that that that's all* **these results**, *that they're actually doing these results, without do* **the experience of this film inside**? |
| Predicted (Cosine) | Why are we not simple robots that *all that* data and produce these *data* **without the experience of this film inside**? |
| Predicted (MaxMargin) | Why aren't we just simple robots that have all this data, make these results, without *making* **the experience of this inside** movie? |
| Predicted (NLLvMF$_{reg}$) | Why are we not simple robots that treat all this data, produce **these results**, without **having the experience of this inside film**? |

Table 11: Example of fluency errors in the baseline model. Red and blue colors highlight translation errors; red are bad and blue are outputs that are good translations, but are considered as errors by the BLEU metric.

