# OpenReview forum: "Von Mises-Fisher Loss for Training Sequence to Sequence Models with Continuous Outputs"
_ICLR.cc/2019/Conference_

### Official Review · AnonReviewer2 · 2018-11-02
**I have some concerns about this paper.**

**Rating:** 6
**Confidence:** 5

**Review:**



[clarity]
This paper is basically well written.
The motivation is clear and reasonable.
However, I have some points that I need to confirm for review (Please see the significance part).


[originality]
The idea of taking advantage of von Mises-Fisher distributions is not novel in the context of DL/DNN research community.
E.g.,
von Mises-Fisher Mixture Model-based Deep learning: Application to Face Verification.

However, as described in the paper, the incorporation of von Mises-Fisher for calculating loss function seems to be novel, to the best of my knowledge.


[significance]
Unfortunately, the experiments in this paper do not fully support the effectiveness of the proposed method.
See below for more detailed comments.


*weak baseline (comparison)
As an anonymous reviewer pointed out, the author should run baseline method with beam search if the authors aim to convince readers (including reviewers) for the effectiveness of the proposed method.
I understand that it is important to investigate the effectiveness of the proposed method in the identical settings. However, it is also important to compare the proposed method with strong baseline to reveal the relative effectiveness of the proposed method comparing with the current state-of-the-art methods.


* open vocabulary setting
I am confused whether the experimental setting for the proposed method is really in an open vocabulary setting or not.
If my understanding is correct, the vocabulary sizes used for the proposed method were 50,000 (iwslt2016) and 300,000 (wmt16), which cannot be an open vocabulary setting.
If this is correct, the applicability of the proposed method is potentially limited comparing with the subword-based approach.
Is there any comment for this question?


* convergence speed
I think the claim of faster convergence of the proposed method in terms of iteration may be misleading. This might be true, but it is empirically proven only by single dataset and single run. The authors should show more empirical results on several datasets or provide a theoretical justification for this claim.


Overall, basically I like the idea of the proposed method.
I also aim to remove the large computational cost of softmax in neural encoder-decoder approach.
In my feeling, the proposed method should be a bit more improved for a recommendation of clear acceptance.

---

> ### Public Comment · (anonymous) · 2018-11-06
> **Some concerns**
>
> I also like the idea of the proposed method as well and have some concerns about the following points.
>
> * Loss function
>
> In the NLLvMF formula following the equation (2), the first term (normalization term) seems to be quite larger than the second term. Let m = 300 as described and k = 15 (empirically average norm of an activated vector with 300D), the loss would be about 440 and 10 for first and second term representatively. This may lead hardness to train in mini-batch.
>
> * Approximated ratio of Bessel function
>
> The approximated ratio of Bessel function suggested in the appendix is different from that in Ruiz-Antoln & Segura, 2016 that proposed {I_v} / {I_(v-1)} approximatation, not {I_(v+1)} / {I_(v-1)}

---

> > ### Author Response · Authors · 2018-11-12
> > **Corrected the typo in the paper**
> >
> > Thank you for your feedback!
> >
> > Here are our responses:
> > 1. While these numbers are accurate, we didn’t face any problems during our mini-batch training. Could you elaborate on why do you think this could lead to hardness during training?
> > 2. This was a typo in the paper, we have corrected it. Thanks for pointing it out!

---

> ### Author Response · Authors · 2018-11-12
> **Response to AnonReviewer2**
>
> Thank you for your detailed feedback. Here are our responses to your comments:
>
> Weak Baseline: As you pointed out, we show results for only greedy decoding to investigate its effectiveness in identical settings. We have since updated the paper with beam search results with the baseline model. The translation quality in our models is still on par or only slightly lower than the results with beam search.
>
> Closed Vocabulary: In IWSLT2016 datasets, the target vocabulary size is around 55000, and around 800,000 in WMT16. The vocabulary sizes we have chosen are not arbitrary but reflect the overlap of the target vocabulary with the pre-trained embedding vocabulary. In principle, it is possible to train the embeddings on a larger monolingual corpora to increase the overlap. But the words for which embeddings could not be found in the embedding table are likely very rare words such as named entities which we handle by using a copy mechanism. Moreover, although subword methods theoretically gives us open-vocabulary setting, they still perform poorly which translating such rare words (see Table 5) because it breaks those rare words into very small units that lose meaning.
>
> Convergence Speed: Due to space limitations, we only report the convergence speed (Figure 1) on one dataset. But the reported results are generated and averaged from multiple runs, and we have achieved consistent performance on all the datasets. We have updated those figures in the appendix. We also report total training time for all the datasets in Table 3 which are also averaged results across multiple runs.

---

### Official Review · AnonReviewer1 · 2018-11-02
**Neat idea backed by a solid technical contribution**

**Rating:** 7
**Confidence:** 4

**Review:**

This paper describes a technique for replacing the softmax layer in sequence-to-sequence models with one that attempts to predict a continuous word embedding, which will then be mapped into a (potentially huge) pre-trained embedding vector via nearest neighbor search. The obvious choice for building a loss around such a prediction (squared error) is shown to be inappropriate empirically, and instead a von Mises-Fisher loss is proposed. Experiments conducted on small-data, small-model, greedy-search German->English, French->English and English->French scenarios demonstrate translation quality on par with BPE, and superior performance to a number of other continuous vector losses. They also provide convincing arguments that this new objective is more efficient in terms of both time and number of learned parameters.

This is a nice innovation for sequence-to-sequence modeling. The technical contribution required to make it work is non-trivial, and the authors have demonstrated promising results on a small system. I’m not sure whether this has any chance of supplanting BPE as the go-to solution for large vocabulary models, but I think it’s very healthy to add this method to the discussion.

Other than the aforementioned small baseline systems, this paper has few issues, so I’ll take some of my usual ‘problems with the paper’ space to discuss some downsides with this method. First: the need to use pre-trained word embeddings may be a step backward. It’s always a little scary to introduce more steps into the pipeline, and it’s uncomfortable to hear the authors state that they may be able to improve performance by changing the word embedding objective. As we move to large training sets, having pre-trained embeddings is likely to stop being an advantage and start being a hindrance. Second: though this can drastically increase vocabulary sizes, it is still a closed vocabulary model, which is a weakness when compared to BPE (though I suppose you could do both).

Smaller issues:

First paragraph after equation (1): “the hidden state … t, h.” -> “the hidden state h … t.”

Equation (2): it might help your readers to spell out how setting \kappa to ||\hat{e}|| allows you to ignore the unit-norm assumption of \mu.

“the negative log-likelihood of the vMF…” - missing capital

Unnumbered equation immediately before “Regularization of NLLvMF”: C_m||\hat{e}|| is missing round brackets around ||\hat{e}|| to make it an argument of the C_m function.

Is predicting the word vector whose target embedding has the highest value of vMF probability any more expensive than nearest neighbor search? Does it preclude the use of very fast nearest neighbor searches?

It might be a good idea to make it clear in 4.3 that you see an extension to beam search for your method to be non-trivial (and that you aren’t simply leaving out beam search for comparability to the various empirical loss functions). This didn’t become clear to me until the Future Work section.

In Table 5, I don’t fully understand F1 in terms of word-level translation accuracy. Recall is easy to understand (does the reference word appear in the system output?) but precision is harder to conceptualize. It might help to define the metric more carefully.

---

> ### Author Response · Authors · 2018-11-12
> **Response to AnonReviewer1**
>
> Thank you for your thorough feedback, we have updated the paper addressing your comments!
>
> Here are our replies to address your comments:
>
> Small Dataset: We show extensive analysis on smaller machine translation datasets (IWSLT) because they take short time to train and hence easier to experiment with. But with our best model we show results on par with softmax based baselines on a much larger WMT German to English dataset with 4.5 million training instances showing the effectiveness of our proposed model in a much broader setting
>
> Pre-trained Embeddings being a hindrance in large datasets: We share your concern on this matter. One of our ongoing projects involves being able to update these output embeddings as part of the training as well. It is possible to do this directly with max-margin loss (which is a contrastive loss) by making the output embeddings trainable, but with other (pairwise) losses, it’ll lead to a degenerate solution (with  all outputs as zeroes). We are currently exploring a wake-sleep-like algorithm to tackle this problem.
>
> Closed vocabulary: This is a good point but the vocabulary can always be increased by training the embeddings on a larger monolingual corpora. Additionally, the words which wouldn’t exist in the vocabulary are most likely (1) proper nouns like named entities which can be handled by the copy mechanism we used in the paper, or (2) rare words. For the latter, although theoretically BPE allows open vocabulary decoding, in practice we see that our model performs much better than BPE baselines in particular on rare words (Table 5). It would be interesting to explore a combination of BPE and our proposed model in future work.
>
> No, Predicting the word with highest probability using vMF has the same computational complexity as nearest neighbor search
>
> We choose F1 to control for noise in the predicted sentence. Recall will only measure a reference word is produced. But by including precision, we measure what fraction of the predicted words are actually in reference. So a sentence producing all the words in reference but also a lot of garbage will be given less score.

---

### Official Review · AnonReviewer3 · 2018-11-05
**cool new approach with some limitations**

**Rating:** 6
**Confidence:** 4

**Review:**

This paper proposes to replace the softmax over the vocab in the decoder with a single embedding layer using the Von Mises-Fisher distribution, which speeds up training 2.5x compared to a standard softmax+cross entropy decoder. The goal is admirable, as the softmax during training is a huge time sink (the proposed approach does not speed up inference due to requiring a nearest neighbor computation over the whole vocab). The approach is evaluated on machine translation (De/F>En and En>F), and the results indicate that there is minor quality loss (measured by BLEU) when using vMF. One huge limitation of the approach is the lack of a beam search-like algorithm; as such, the model is compared to greedy softmax+CE decoders (I would like to see numbers with a standard beam search model as well just to emphasize the quality drop from the state-of-the-art systems). With that said, I found this approach quite exciting and it has potential to be further improved, so I'm a weak accept.

comments:
- is convergence time the right thing to measure when you're comparing the two different types of models? i'd like to see something like flops as in the transformer paper.
- relatedly, it's great that you can use a bigger batch size! this could be very important especially for non-MT tasks that require producing longer output sequences (e.g., summarization).
- it looks like the choice of pretrained embedding makes a very significant difference in BLEU. i wonder if contextualized embeddings such as ELMo or CoVE could be somehow incorporated into this framework, since they generally outperform static word embeddings.

---

> ### Author Response · Authors · 2018-11-12
> **Response to AnonReviewer3**
>
> Thank you for your feedback! Here are our responses:
>
> Beam Search: As pointed in one of our earlier comments (https://openreview.net/forum?id=rJlDnoA5Y7&noteId=HkxNHeoR57), beam search is not impossible to do but with our proposed model but is not trivial as to just using k nearest neighbors as candidates. It requires substantial investigation due to which we leave it as future work. We included BLEU scores with just greedy search for fair comparison with baselines and keeping in line with earlier work with similar motivation to ours (https://arxiv.org/abs/1704.06918). But thank you for your suggestion, we have now updated the draft to include results with beam search as well. Note that translation quality in our models is still on par or only slightly lower than the results with beam search.
>
> Since the beam search is known to slow down decoding and there has been work in the past to get rid of it from softmax based architectures (https://openreview.net/forum?id=rJZlKFkvM, https://arxiv.org/pdf/1701.02854.pdf). The latter paper, for example, proposes a deterministic alternative to decoding where instead of sampling from softmax output at each step, you feed the entire softmax distribution to the next step. We plan to explore a similar approach in the future where the output vector is fed directly to the next step as opposed to finding it’s nearest neighbor and feeding that to the next step.
>
> Convergence Time: Convergence time, which is reflective of total training time is crucial factor in machine translation systems some of which can take weeks to train and is reported by the transformer paper as well (https://arxiv.org/abs/1706.03762). We report number of samples processed per second (Figure 1) instead of FLOPs (floating point operations) per second, as was reported in the transformer paper. The metrics correlate. We believe that FLOPs measure is more noisy because it's hard to keep GPUs utilized at 100%.
> Use of ELMo or CoVe: This is a great suggestion, thank you. But as you pointed out, they are contextual embeddings and it’s not clear how to directly incorporate them. But it would an exciting future direction for this work.

---

### Public Comment · (anonymous) · 2018-10-02
**Weak baseline models**

1. A big advantage of the proposed method is using pre-trained embeddings on large monolingual corpora. This gives "substantial improvements over softmax and BPE baselines in translating less frequent and rare words". Pre-trained word embeddings are also useful for standard NMT ( http://aclweb.org/anthology/N18-2084.pdf ) but the authors failed to leveraging pre-trained embeddings for baseline models. BTW, embeddings could also be trained on BPEed corpora.
2. The nearest-neighbor decoding should support K-NN in principle. Why couldn't beam search be applied to the proposed method? Not using beam search also made the baselines weak.

---

> ### Public Comment · ~Tzu-Hsiang_Lin1 · 2018-10-04
> **Not so weak**
>
> 1. Your link is broken.  The author have both results on whether the target embeddings is tied or not. (See Section 4.4 and Table 1.)
> 2. The reason why Beam Search cannot be directly applied is because Von Mises-Fisher loss is not a probability, and Beam search calculates the probability of a sequence.
>
> The main idea here is a new objective(directly predicting word embeddings) with a loss function(Von Mises-Fisher) that works.   As long as other settings are the same, the baselines should be reasonable enough.

---

> > ### Public Comment · (anonymous) · 2018-10-05
> > **not convinced**
> >
> > 1. The link is fixed.
> > 2. We can always choose to tie or not tie target embeddings for the standard NMT. How does it related to embedding pre-training? The target input/output embeddings in (Press & Wolf, 2016) refer to the matrix of size (H × V) in this paper, but the pre-trained embeddings have the size (m × V). Anyway, the proposed method leverages the knowledge of large monolingual corpora while baselines don't (but they could).
> > 3. Do you mean the lower scores with no-tying embeddings are fairer to be compared?
> > 4. You confirmed that the proposed method cannot use beam search, which is a disadvantage. It's not fair to compare with baselines with restricted ability -- using beam search.
> > 5. Yes, this paper presents an interesting idea of directly predicting word embeddings accompanied with the Von Mises-Fisher loss function. But it has not been proved to be effective -- on par with or better than fair baselines.

---

> ### Author Response · Authors · 2018-10-12
> **Responses and Clarifications**
>
> Thank you for your feedback. Based on your comments, here are our responses:
>
> 1) Initializing Embeddings: Following your comment, we have conducted experiments with initializing the embeddings in softmax based models. Our model still performs on par with those baselines: for example, for fr-en setup, initializing the embeddings gives a small gain of 0.2 BLEU, which is still in line with reported results. We'll update these results in the draft.
>
> 2) Using BPE Embeddings: We have pointed out in footnote 11 that for different language pairs, different number BPE merge operations are often used. Moreover, the BPE operations are performed by using training data from both languages. This will require different target embeddings to be trained for different language pairs increasing the total training time per language pair.
>
> 3) Decoding with Beam Search: In principle, it is possible to generate candidates for beam search as you pointed out by using K-Nearest Neighbors. But how to rank the partially generated sequences is not trivial (one could use the loss values themselves to rank, but initial experiments with this setting didn't result in significant gains). In this work, we focus on enabling training with continuous outputs efficiently and accurately giving us huge gains in training time. The question of decoding with beam search requires substantial investigation and we leave it as future work. This is in line with recent NMT work with similar motivation of alleviating softmax bottleneck problem (https://arxiv.org/pdf/1704.06918.pdf) who also do best-first decoding.
> It is noteworthy that beam search is not the only way to improve decoding quality with the proposed architecture. For example: this paper (https://arxiv.org/pdf/1701.02854.pdf) proposes a deterministic alternative to decoding where instead of sampling from softmax output at each step, you feed the entire softmax distribution to the next step. There are perhaps other ways of decoding which could be explored in future work, beam search is not the only option.

---

### Public Comment · (anonymous) · 2018-10-13
**You may want to consider Adaptive softmax for comparison**

Have you tried Adaptive softmax? It typically reduces the training time as well.

---

> ### Author Response · Authors · 2018-10-19
> **Training time isn't the only issue we tackle in this work**
>
> Thank you for you response.
>
> Adaptive softmax can be categorized into structural approximations of softmax. We will update it in our background section (section 2). While it would achieve good gains in terms of training time over "word based" softmax, this gain would not be significant over BPE based models which already have a very small vocabulary. Moreover, "word based" softmax models don't perform on par with SOTA in many MT systems (see Table 2).
>
> Our main focus in this work was to get rid of the softmax layer which will also likely help in other tasks related to language generation. And we compared with MT baselines (BPE based) which are strong both in terms of speed and accuracy

---

### Meta-Review · Area_Chair1 · 2018-12-14
**weak accept**

**Confidence:** 4
**Recommendation:** Accept (Poster)

**Metareview:**

this is a meta-review with the recommendation, but i will ultimately leave the final call to the programme chairs, as this submission has a number of valid concerns.

the proposed approach is one of the early, principled one to using (fixed) dense vectors for computing the predictive probability without resorting to softmax, that scales better than and work almost as well as softmax in neural sequence modelling. the reviewers as well as public commentators have noticed some (potentially significant) short comings, such as instability of learning due to numerical precision and the inability of using beam search (perhaps due to the sub-optimal calibration of probabilities under vMF.) however, i believe these two issues should be addressed as separate follow-up work not necessarily by the authors themselves but by a broader community who would find this approach appealing for their own work, which would only be possible if the authors presented this work and had a chance to discuss it with the community at the conference. therefore, i recommend it be accepted.